# The Relationship between Motor Coordination Ability, Cognitive Ability, and Academic Achievement in Japanese Children with Autism Spectrum Disorder and Attention Deficit/Hyperactivity Disorder

**DOI:** 10.3390/brainsci12050674

**Published:** 2022-05-21

**Authors:** Takuya Higashionna, Ryoichiro Iwanaga, Akiko Tokunaga, Akio Nakai, Koji Tanaka, Goro Tanaka

**Affiliations:** 1Department of Rehabilitation, Faculty of Health Sciences, Tokyo Kasei University, Inariyama, Sayama 350-1398, Saitama, Japan; higashionna-t@tokyo-kasei.ac.jp; 2Department of Occupational Therapy Sciences, Nagasaki University Graduate School of Biomedical Sciences, Sakamoto, Nagasaki 852-8102, Nagasaki, Japan; akiko0923@nagasaki-u.ac.jp (A.T.); goro@nagasaki-u.ac.jp (G.T.); 3The Center for the Study of Child Development, Graduate School of Clinical Education, Institute for Education, Mukogawa Women’s University, Nishinomiya 663-8558, Hyogo, Japan; anakai.kodomo@gmail.com; 4Department of Rehabilitation Sciences, Gunma University Graduate School of Health Sciences, Showa, Maebashi 371-8514, Gunma, Japan; kojit929@gunma-u.ac.jp

**Keywords:** motor coordination, cognitive ability, academic achievement, autism spectrum disorder, attention deficit/hyperactivity disorder

## Abstract

Motor coordination abilities are related to cognitive abilities and academic achievement in children with neurodevelopmental disorders. However, the similarities and differences of these relationships in children with autism spectrum disorder (ASD) and attention deficit/hyperactivity disorder (AD/HD) have not been explored. The purpose of this study was to investigate the relationship between motor coordination abilities, cognitive abilities, and academic achievement in Japanese children with ASD and AD/HD. Participants included 20 children with ASD, 20 children with AD/HD, and 20 typically developing children, matched for age and gender. Their motor coordination abilities were assessed with the Movement Assessment Battery for Children-2 (MABC-2). Furthermore, cognitive ability and academic achievement were assessed with the Kauffman Assessment Battery for Children-II (K-ABCII). Results demonstrated that the MABC-2 Total score significantly correlated with the K-ABCII Simultaneous processing, Planning, Total cognitive ability, Writing and Arithmetic scores in children with ASD. However, in children with AD/HD, there was no significant correlation between MABC-2 and K-ABCII subscale scores. The results of this study indicated that the relationship between motor coordination ability, cognitive ability, and academic achievement differs between ASD and AD/HD. This difference might indicate the non-similarity of neurological characteristics and encourage consideration for an approach that accommodates the features of neurodevelopmental disorders in children.

## 1. Introduction

Motor coordination impairment are highly presented in children with neurodevelopmental disorders. Among them, autism spectrum disorder (ASD), attention deficit/hyperactivity disorder (AD/HD), and specific learning disorder (SLD) are known as comorbidities of developmental coordination disorder (DCD) [1]. In a population-based study in Hirosaki city in Japan, it was revealed that 63.2% of children with ASD have co-occurring DCD [2]. Another population-based study that using a population sample of 7-year-old children in Swedish reported 47% of children with AD/HD diagnosed with DCD [3]. Thus, motor impairments are being recognized as an important part of the features of children with neurodevelopmental disorders.

Motor coordination consists of psychological processes such as sensory input, perceptual processing, cognitive processing, and action production. Recent research has revealed that there were relations between motor and cognitive skills in typically developing children [4]. Furthermore, previous study has demonstrated that children with DCD show decreased cognitive abilities [5].

Recently, the relationship between motor coordination ability and academic achievement also has been investigated. A systematic review published by Macdonald et al. identified that there are significant associations between motor proficiency and academic performances in typically developing children [6]. Previous study of school-aged children using regression analysis also found that motor coordination was a predictor of positive academic performance [7]. Our previous study also revealed that the Manual Dexterity and Balance scores of the Movement Assessment Battery for Children-2 (MABC-2) were significantly correlated with the total academic achievement score of the Kauffman Assessment Battery for Children-II (K-ABCII) in Japanese children with neurodevelopmental disorders [8]. Furthermore, previous studies have revealed that children with DCD showed significantly poorer performance in reading, writing, spelling, literacy and numeracy [9] and mathematics [10]. Thus, it can be concluded that motor coordination abilities are related to academic achievement.

However, no study have investigated the relationship between motor coordination ability, cognitive ability and academic achievement in children with ASD and AD/HD despite recognizing the importance of motor coordination problems among such children. Although it is known that ASD and AD/HD frequently co-occur each other, comparative meta-analyses reported that the brain structural and functional abnormalities were different between ASD and AD/HD [11]. Thus, it is estimated that the relationship between motor coordination ability, cognitive ability and academic achievement were also different between ASD and AD/HD. Furthermore, investigating these relationships in children with ASD and AD/HD might clarify the similarities and differences among movement characteristics between both diagnosis groups. Thus, we conducted a study examining the relationships between motor coordination ability, cognitive ability, and academic achievement in Japanese children with ASD and AD/HD and whether there are any similarities and differences between these groups.

## 2. Materials and Methods

### 2.1. Participants

The participants in this study consisted of 60 school-aged (6–12 years old) children: 20 children with ASD (8.6 ± 2.1 years old), 20 children with AD/HD (8.8 ± 1.6 years old) and 20 typically developing (TD) children (8.4 ± 1.4 years old). The male-female ratio of all groups was 17:3. The three groups did not statistically differ in age.

Children with ASD and AD/HD were recruited by purposeful sampling through occupational therapists, speech therapists, and physicians in medical and welfare centers for developmental disorders and through parents’ associations for children with neurodevelopmental disorders and primary schools in Nagasaki city in Japan. The children were provided with special support services in resource rooms in each school. All children were diagnosed with neurodevelopmental disorders by a pediatrician based on the DSM-5 criteria (20 ASD and 20 AD/HD). In this study, there were two children with ASD for whom AD/HD was also suspected and two children with AD/HD for whom ASD was also suspected. For each case, we eliminated the suspected diagnosis. Intellectual ability of children with ASD and AD/HD was evaluated by using the Wechsler Intelligence Scale for Children-IV (WISC-IV). All children had a full-scale IQ (FSIQ) greater than 70, and the average FSIQ of the ASD group (98.9 ± 19.5) and AD/HD group (91.4 ± 12.8) did not statistically differ. In this study, children who had been diagnosed with any physical or learning disability (i.e., SLD) were excluded.

Further, TD children were also recruited from primary schools in the same region. Inclusion criteria for TD children were no intellectual impairment, no learning difficulties, no behavioral problems, and no diagnosed emotional, neurological, or motor disorders.

### 2.2. Procedures

At first, parental informed consent was obtained for each child. All children were tested individually in separate rooms by researchers. In addition, children both in the ASD group and AD/HD group were administered the K-ABCII. Parents of the children in both groups were given the Autism Spectrum Screening Questionnaire (ASSQ), Pervasive Developmental Disorders Autism Society Japan Rating Scale (PARS-TR) and the ADHD Rating Scale-IV (ADHD RS-IV) and completed.

### 2.3. Instruments

#### 2.3.1. Movement Assessment Battery for Children-2 (MABC-2)

The MABC-2 was administered to assess motor coordination impairments. The standardized MABC-2 test has three age bands, covering 3–6, 7–10, and 11–16 years, and its total duration is 20–40 min. Moreover, the MABC-2 test consists of eight tasks and the results of which provide a total score and three component scores: Manual Dexterity (MD), Aiming and Catching (AC), and Balance. The raw score for each task is converted to a standard score by using the test manual, and standard scores for each component can be calculated. The sum of the eight task scores can be recorded as a total standard score. Standard scores have a mean of 10 and a standard deviation (SD) of 3. Higher standard scores indicate greater motor coordination ability. Since the standardization of the Japanese version of the MABC-2 is currently in progress [12,13], we employed original data from the UK for calculations of the standard scores [14]. The validity and reliability of the test were confirmed using the test manual.

#### 2.3.2. The Japanese Version of the Kauffman Assessment Battery for Children-Second Edition (K-ABCII)

The K-ABCII was administered to assess cognitive abilities and academic achievement [15]. This standardized K-ABCII test of American version was developed in 2004, and the Japanese standardized version was developed in 2013. The Japanese version of the K-ABCII covers from 2 years 6 months to 18 years 11 months and includes cognitive and academic achievement tests. These tests are based on the Kaufman model, which is based on Luria’s theory of brain functioning. The Kaufman model allows one to measure eight abilities, with four subscales measuring cognitive abilities (Sequential Processing, Simultaneous Processing, Planning, and Learning) and four subscales measuring academic achievement (Knowledge, Reading, Writing, and Arithmetic). The basic test for all ages includes 20 subtests (11 cognitive and 9 academic). The results yield standard scores based on the total scores, converted into percentiles, for each test item. The standard score is a mean score of 100 with an SD of 15. Higher values indicate higher cognitive ability and academic achievement.

#### 2.3.3. Autism Spectrum Screening Questionnaire (ASSQ)

The ASSQ was used to screen for ASD and was the first instrument developed as a screening for Asperger Syndrome. However, it was later relabeled the Autism Spectrum Screening Questionnaire as a screening tool for other ASDs as well [16]. It has parent and teacher forms and consists of 27 items that are measured on a three-point scale: “not true” (0), “sometimes true” (1), and “certainly true” (2). The range of the total ASSQ score is 0 to 54. Cut-off scores of 22 for teachers’ ratings and 19 for parents’ ratings were recommended for screening to identify high-functioning children with ASDs in clinical settings.

#### 2.3.4. Pervasive Developmental Disorders Autism Society Japan Rating Scale (PARS-TR)

The PARS-TR was used to screen for ASD. It is an instrument developed and published in Japan [17], and involves the evaluation of symptoms of pervasive developmental disorders (PDDs). The PARS-TR is used through a semi-structured interview conducted with a parent or family member of child. It consists of 34 items about the typical behavioral symptoms of PDDs on three levels: “none” (0), “somewhat apparent” (1), and “apparent” (2). The time required to implement the PARS-TR remains between 30–90 min, depending on the interviewer’s proficiency and the age and symptoms of child [17].

#### 2.3.5. ADHD Rating Scale-IV (ADHD RS-IV)

The ADHD RS-IV was used to screen for AD/HD. We used the parent version. The ADHD RS-IV consists of 18 items and 2 subscales (inattention and hyperactivity-impulsivity), each containing 9 items. Parents were required to rate the frequency of each of the ADHD symptoms that occurred over the previous 6 months on a 5-point Likert scale that ranged from 0 to 4: “never” (0), “rarely” (1), “sometimes” (2), “often” (3), and “very often” (4). The sum of all the scores for the 18 items results in a total score that ranges from 0 to 54.

### 2.4. Statistical Analysis

The software IBM SPSS Statistics version 22.0 (IBM Corp, Armonk, NY, USA) was used for data analysis, and the significance level was set to *p* < 0.05. We used the *t*-test to analyze differences in the questionnaire (ASSQ, PARS-TR, and ADHD RS-IV) scores and K-ABCII scores between the ASD group and AD/HD group.

One-way ANOVA with the Tukey HSD post-hoc test was used to compare the MABC-2 standard scores of the children with ASD and AD/HD and the TD children.

We used Pearson’s correlation analysis to examine the relationship between the MABC-2 and cognitive ability scores of the K-ABCII. Pearson’s correlation analysis was also used to examine whether the MABC-2 and K-ABCII scores were related to intellectual ability in the ASD group and AD/HD group, and to examine the FSIQ’s relation to the MABC-2 MD and K-ABCII academic achievement test scores. Additionally, we used partial correlation analysis to examine the relationship between the standard scores of the MABC-2 and the academic achievement test of the K-ABCII in the ASD group and AD/HD group while controlling FSIQ.

## 3. Results

Table 1 shows the summary results of the questionnaire scores and the mean standard scores and SDs of the K-ABCII in the ASD group and AD/HD group.

### 3.1. Motor Coordination Ability

Table 2 shows the results of the one-way ANOVA for standard scores of the MABC-2.

The ASD group and AD/HD group had significantly lower MABC-2 Total scores than the TD children (*p* < 0.05). Moreover, the ASD group had significantly lower MABC-2 scores for all components (MD, AC, and Balance) than the TD children. The AD/HD group also had significantly lower MABC-2 Balance scores than the TD children. There were no significant differences in the MABC-2 scores between the ASD group and the AD/HD group.

### 3.2. Correlation between Motor Coordination Ability and Cognitive Ability/Academic Achievement

Table 3 list the correlations between the standard scores of the MABC-2 and cognitive ability scores of the K-ABCII in the ASD group and AD/HD group. In the children with ASD, the Total score of the MABC-2 was significantly correlated with the K-ABCII Simultaneous processing (r = 0.55, *p* = 0.011), Planning (r = 0.55, *p* = 0.012), and Total cognitive ability (r = 0.49, *p* = 0.030) scores. Moreover, the MABC-2 MD score was significantly correlated with the Simultaneous processing (r = 0.58, *p* = 0.007), Planning (r = 0.65, *p* = 0.002), and Total cognitive ability (r = 0.51, *p* = 0.023) scores. In children with AD/HD, there was no significant correlation.

Table 4 list the correlations between the standard scores of the MABC-2 and the academic achievement test of the K-ABCII. In the ASD group, the MABC-2 Total score had significantly positive correlations with the K-ABCII Writing (r = 0.75, *p* = 0.003) and Arithmetic (r = 0.67, *p* = 0.012) scores, and the MABC-2 MD score was significantly correlated with the K-ABCII Arithmetic scores (r = 0.59, *p* = 0.032). Moreover, the MABC-2 Balance score was also significantly correlated with the K-ABCII Writing (r = 0.75, *p* = 0.003) and Arithmetic (r = 0.61, *p* = 0.028) scores. In the AD/HD group, there was no significant correlation between the standard scores of the MABC-2 and the academic achievement test of the K-ABCII.

## 4. Discussion

### 4.1. Motor Coordination Ability

First, the MABC-2 scores of the ASD group and AD/HD group were compared to those of the age-matched TD children to identify motor coordination impairments in this study. The results of this comparison showed that both the ASD group and AD/HD group had significantly lower the MABC-2 Total scores than the TD children. This result is consistent with previous studies showing significant motor impairments in children with ASD and AD/HD [18,19,20,21]. Furthermore, it was revealed that the MABC-2 component scores which significantly lower than TD children differs between the ASD and AD/HD group. In this study, although all component scores of the MABC-2 significantly lower than TD children in the ASD group, only the Balance score of the MABC-2 significantly lower than TD children in the AD/HD group. Therefore, it is suggested that the feature of motor coordination impairment differ between ASD and AD/HD.

### 4.2. Correlation between Motor Coordination Ability and Cognitive Ability

The results of the analysis examining the relationship between motor coordination and cognitive ability showed a significant correlation between the MABC-2 Total score and the K-ABCII Total cognitive score in children with ASD, but there was not significant correlation in children with AD/HD. Therefore, it is suggested that the relationship between motor coordination ability and cognitive ability is closer in children with ASD than for those whom AD/HD. Especially, the MABC-2 MD and Total scores were significantly correlated with the K-ABCII Simultaneous Processing and Planning scores in children with ASD. Therefore, it is suggested that motor coordination abilities, especially fine motor abilities are related to simultaneous processing and planning ability of cognitive abilities.

According to the K-ABCII manual [15], Simultaneous Processing is the ability to combine and process large amounts of information at once, and this ability is required for integrating spatial stimuli and solving problems efficiently. Recent research indicated potential deficits in the simultaneous processing of multiple inputs and outputs in children with ASD and suggested that deficits in simultaneous processing have significance beyond motor function in ASD [22]. Therefore, results of this study are consistent with findings of previous studies and reveal that simultaneous processing ability is related to fine motor coordination ability in children with ASD.

In the meanwhile, Planning of the K-ABCII is an ability that is required for hypothesizing, flexibility, and impulse control. Previous studies have reported the issues with praxis and executive function (EF) in ASD [23,24,25]. Dyspraxia, a dysfunction of praxis, is identified as a DCD, and children with ASD are reported to have general praxis issues relating to not only physical imitations but also making gestures for instructions and operating objects [26]. The relationship between praxis and basic motor skills has been examined, and its effect on motor coordination has been shown [27]. Moreover, EF is a term that includes the process of controlling emotions, which is necessary to maintain an action to achieve a goal effectively through physical, cognitive, and emotional self-control [28]. This includes functions such as inhibition, working memory, cognitive flexibility, planning and fluency. Previous studies on EF have indicated its relation to motor functions [29]. Furthermore, executive functioning disorders are found among children with neurodevelopmental disabilities such as ASD and AD/HD, although symptoms differ among the two [30], and children with ASD are indicated to have relatively pervasive executive functioning disorders compared to children with AD/HD [25]. Based on the above, it is suggested that the pathognomonic cognitive function disorders such as praxis and EF are indicated to affect motor coordination in children with ASD. Thus, it is important to consider motor coordination impairment including the association of cognitive abilities in children with ASD for assessment and intervention.

### 4.3. Correlation between Motor Coordination Ability and Academic Achievement

In our analysis of the relationship between motor coordination ability and academic achievement, the MABC-2 Total score was significantly correlated with the K-ABCII Writing and Arithmetic scores in children with ASD. Previous studies have reported weak reading comprehension, mathematics skills, and interpretive skills among children with ASD [31,32]. However, no study has directly investigated the relationship between motor coordination ability and academic achievement in children with ASD and AD/HD. Our study clarified for the first time that the acquisition of writing and mathematics among children with ASD is related to the MABC-2 Total score. The MABC-2 MD score was significantly correlated with the Arithmetic score, and the MABC-2 Balance score was significantly correlated with Writing and Arithmetic scores in children with ASD. Regarding the MABC-2 MD, our previous study revealed that fine motor skills are associated with a wide range of academic achievements (e.g., in language, reading, writing, and arithmetic) in children with neurodevelopmental disorders [8]. Therefore, it is suggested that, in children with ASD, there is a close relationship between fine motor skills and arithmetic skills. Regarding MABC-2 Balance, this study revealed that balance skill is significantly correlated with writing and arithmetic skills in children with ASD. Several factors may account for this result. In children with ASD, it is suggested that there are cognitive features (i.e., praxis, EF) consisting of motor coordination and academic skills. Moreover, it is possible that balance and academic skills share a common neural basis. Balance, as a motor function, is associated with the cerebellum. Some neuroimaging studies have revealed that the structure of cerebellum plays an important role in not only motor control but also non-motor functions such as language, learning, and memory [33,34]. Furthermore, previous studies revealed that cerebellar lesions are important factors in dyslexia and dyscalculia [35]. Therefore, the results of this study suggest a common neural basis for balance, cognitive functions, and academic skills.

On the contrary, there was not a significant correlation between the MABC-2 and K-ABCII in the AD/HD group. In the study of latent neuroanatomic variables in children with AD/HD, the skill of aiming and catching was significantly linked to latent variables for the premotor/motor cortical region and superior cerebellar lobules. However, these links were not moderated by the severity of symptoms of inattention, hyperactivity and impulsivity [36]. Thus, the neuroanatomic basis affecting motor coordination problems might not be related to the severity of AD/HD symptoms. However, relationships between AD/HD symptoms and academic skills were reported [37,38]. Therefore, neuroanatomic dysfunction, which is related to motor coordination problems in children with AD/HD, may not be related to AD/HD symptoms, which affect academic skills.

### 4.4. Limitations

Finally, the limitations of this study should be considered. First, we did not have access to information about TD children’s IQ and K-ABCII data. Previous systematic reviews showed that there are relationships between motor, cognitive abilities, and academic achievement in TD children [4,6]. Thus, it is necessary for future study to investigate these relationships including TD children. Future studies may provide increased understanding the characteristics of motor coordination abilities in children with ASD and AD/HD and clarify the similarities and differences neurological bases between diagnosis groups. Second, there was no significant difference between ASD and AD/HD in ADHD RS-IV scores, and the AD/HD group had a PARS-TR score above the cut-off. Previous studies reported that AD/HD symptoms occur frequently in children with ASD. Thus, although there were no cases of children diagnosed with both ASD and AD/HD in this study, it is possible that there were children in this study who had symptoms of both. Moreover, the methodology used to select the patients and to establish diagnosis wasn’t enough. Even if clinical examination may be valuable in clinical setting, it should be associated with validated and semi-structured instruments in recruiting patients for purpose of this study. Finally, the sample size for this study was small. Further research should be conducted with larger samples.

## 5. Conclusions

The present study confirmed that the relationship between motor coordination ability, cognitive ability, and academic achievement differs for ASD and AD/HD. Furthermore, these correlations were more significant among children with ASD compared to those with AD/HD. This could be affected by the pathognomonic symptoms of ASD, such as issues related to praxis and EF. For assessment and interventions, current findings emphasized the importance of considering and accommodating the features of neurodevelopmental disorders in children.

## Figures and Tables

**Table 1 brainsci-12-00674-t001:** Comparison of scores for questionnaire and K-ABCII between the ASD group and AD/HD group.

	ASD (*n* = 20) Mean (SD)	AD/HD (*n* = 20) Mean (SD)	t	*p*
ASSQ	22.0 (10.5)	15.0 (8.8)	2.28	0.03 *
PARS-TR	23.1 (11.1)	15.1 (8.9)	2.48	0.02 *
ADHD RS-IV	22.0 (9.9)	23.9 (9.6)	−0.61	0.55
K-ABCII Cognitive ability				
Sequential processing	96.6 (16.7)	84.8 (12.8)	2.94	<0.01 **
Simultaneous processing	101.0 (16.1)	99.2 (12.2)	0.40	0.69
Planning	105.0 (14.5)	92.6 (15.8)	2.59	<0.01 **
Learning	107.0 (16.3)	103.6 (13.9)	0.72	0.48
Total	102.6 (15.8)	92.0 (13.0)	2.33	0.03 *
K-ABCII Academic achievement				
Knowledge	101.8 (15.2)	94.5 (12.5)	1.65	0.11
Reading	101.6 (18.2)	92.1 (15.0)	1.80	0.08
Writing	97.3 (19.5)	89.1 (18.0)	1.34	0.19
Arithmetic	96.9 (15.9)	93.4 (15.9)	0.70	0.49
Total	99.7 (16.7)	90.3 (16.9)	1.77	0.08

ASSQ, Autism Spectrum Screening Questionnaire; PARS-TR, Pervasive development disorders Autism society japan Rating Scale; ADHD RS-IV, ADHD Rating Scale-IV; K-ABCII, Kauffman Assesssment Battery for Children-Second edition; ASD, autism spectrum disorder; AD/HD, attention deficit/hyperactivity disorder; SD, standard deviation. ** *p* < 0.01; * *p* < 0.05.

**Table 2 brainsci-12-00674-t002:** Comparison of the MABC-2 standard scores between ASD group, AD/HD group, and typically developing children.

	ASD (*n* = 20) Mean (SD)	AD/HD (*n* = 20) Mean (SD)	Typically Developing (*n* = 20) Mean (SD)
Manual Dexterity	9.2 (3.6) ^a^	10.5 (2.7)	12.4 (3.0)
Aiming & Catching	7.3 (3.5) ^a^	9.0 (3.0)	10.8 (3.4)
Balance	10.3 (3.5) ^a^	10.6 (2.6) ^b^	13.5 (1.9)
Total	8.8 (3.4) ^a^	10.2 (2.6) ^b^	13.0 (2.4)

ASD, autism spectrum disorder; AD/HD, attention deficit/hyperactivity disorder; SD, Standard Deviation. ^a^ ASD vs. Typically developing, *p* < 0.05; ^b^ AD/HD vs. Typically developing, *p* < 0.05.

**Table 3 brainsci-12-00674-t003:** Correlations between standard scores of the MABC-2 and cognitive ability test of the K-ABCII.

	Sequential Processing	Simultaneous Processing	Planning	Learning	Total Cognitive Ability
ASD					
Manual Dexterity	0.07	0.58 **	0.65 **	0.32	0.51 *
Aiming & Catching	0.03	0.27	0.10	−0.11	0.11
Balance	−0.02	0.34	0.41	0.36	0.36
Total MABC-2	0.08	0.55 *	0.55 **	0.29	0.49 *
AD/HD					
Manual Dexterity	0.42	0.25	0.34	0.15	0.42
Aiming & Catching	−0.18	0.09	−0.14	0.01	−0.09
Balance	0.26	0.29	−0.08	0.14	0.23
Total MABC-2	0.25	0.15	0.04	0.11	0.20

MABC-2, Movement Assessment Battery for Children-2; K-ABCII, Kauffman Assesssment Battery for Children-Second edition; ASD, autism spectrum disorder; AD/HD, attention deficit/hyperactivity disorder. ** *p* < 0.01; * *p* < 0.05.

**Table 4 brainsci-12-00674-t004:** Correlations between standard scores of the MABC-2 and academic achievement test of the K-ABCII.

	Knowledge	Reading	Writing	Arithmetic	Total Academic Achievement
ASD					
Manual Dexterity	0.26	0.24	0.53	0.59 *	0.52
Aiming & Catching	−0.05	0.09	0.48	0.38	0.26
Balance	−0.34	0.14	0.75 **	0.61 *	0.31
Total MABC-2	−0.03	0.17	0.75 **	0.67 *	0.46
AD/HD					
Manual Dexterity	0.35	0.10	0.33	0.22	0.26
Aiming & Catching	−0.04	−0.29	−0.09	−0.12	−0.18
Balance	0.11	−0.03	0.02	0.05	0.34
Total MABC-2	0.17	−0.04	0.14	0.06	0.07

MABC-2, Movement Assessment Battery for Children-2; K-ABCII, Kauffman Assesssment Battery for Children-Second edition; ASD, autism spectrum disorder; AD/HD, attention deficit/hyperactivity disorder. ** *p* < 0.01; * *p* < 0.05.

## Data Availability

Not applicable.

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
