# Peer review of "The Relationship between Motor Coordination Ability, Cognitive Ability, and Academic Achievement in Japanese Children with Autism Spectrum Disorder and Attention Deficit/Hyperactivity Disorder"

_brainsci, 2022, doi:10.3390/brainsci12050674_

Round 1

Reviewer 1 Report

The authors report additional evidence for a relationship between motor/cognitive difficulties and children with ASD or AD/HD. The topic is of interest, and findings may contribute to stimulate further research. However, the manuscript would benefit strongly with a major revision. In particular, the Introduction and Conclusions would benefit from more additional considerations. Please review entire manuscript for syntax, grammar and word-choice. Specific details by manuscript section are provided below:

-Is there a estimate prevalence of developmental coordination disorder in ASD and AD/HD populations? In Japan or wordwide?

-I suggest improving the Introduction Section, with the aim to be focused on ASD or AD/HD children and their developmental coordination disorders, in a more systematic way and less confused

-Why did the authors refer to “neurological bases” (Lines 74-75’). If the authors suppose that it is an essential concept in these diagnosis groups, they should expand this consideration in the Introduction section

-I suggest describing Procedures (2.3 Section) before Instruments (2.2 Section)

-In the Statistical analysis (2.4 Section), I suggest clarifying “questionnaire scores” (lines 167-168)

-In Conclusion (Section 5) the authors said: “This could be affected by the core symptoms of ASD, such as issues related to praxis and EF”, but they did not discuss any core symptoms of ASD elsewhere. I suggest doing it. The authors did non measured any relationship between motor difficulties and core symptoms in ASD or AD/HD children

Author Response

Dear Reviewer 1,

We are most grateful to you for the helpful comments on the original version of our manuscript entitled “The Relationship between Motor Coordination Ability, Cognitive Ability, and Academic Achievement in Japanese Children with Autism Spectrum Disorder and Attention Deficit/Hyperactivity Disorder”. We have taken all these comments into account and herewith submit revised version of our paper.

We hope that the revised version of our paper is now suitable for publication in Brain Sciences and we look forward to hearing from you at your earliest convenience.

Reviewer 2 Report

The paper reports a study on the relationship between coordination abilities and cognitive abilities/academic achievement in children with Autism Spectrum Disorders (ASD) and Attention Deficit and Hyperactivity Disorder (ADHD).

Actually, it follows and extends a study published previously by the same authors which showed that coordination abilities in children with neurodevelopmental disorders (NDD) are significantly correlated to their cognitive and academic performance.

However, considering neurodevelopmental disorders as a whole obviously lacks precision and the authors therefore aimed at categorizing NDD by comparing the same relationship in two separate groups: ASD and ADHD.

The present study shows that M-ABC scores are correlated to cognitive/academic performance in children with ASD but not significantly in children with ADHD. Authors propose several hypotheses to explain the difference, namely the role of the cerebellum that could be implicated in both the coordination abilities and various cognitive (dys)functions in ASD, and the direct implication of ADHD symptoms, not related to coordination abilities, on the academic achievement.

A better characterization of different NDD categories, by describing and understanding their very specificity, underlying pathogenetic mechanisms, which may be required to offer more specific and well-adjusted interventions is worth and even crucial. Many previous studies aimed at highlighting similarities/differences, on the clinical, neuropsychological and anatomical levels. Undoubtedly the goal pursued by the authors is of great interest, but it is challenging because of the frequent overlap and co-morbidities between these two categories.

From our point of view, the authors partially succeeded in selecting specific phenotypes in both groups. Firstly, the methodology used to select the patients and to establish diagnoses isn’t strong enough, which is a limitation. The diagnosis of ASD relies on parental questionnaires and DSM criteria. Even if clinical examination by trained professionals may be valuable in clinical setting, it should ideally be associated with validated and semi-structured instruments in recruiting patients for research purpose.

Secondly, scores obtained in the ADHD-questionnaires were quite similar in both groups, suggesting that most of ASD patients presented a co-morbid ADHD, which is actually frequent and was acknowledged by the authors in the limitation chapter. These methodological limitations are in line with current debates in the scientific community about the crucial need to study more pure and specific cohorts of patients, especially in the large spectrum of autism, in order to avoid limitations and issues due to extreme heterogeneity and/or symptoms overlap.

The paper is clear, concise and contributes to crucial attempts to explore the overlap/distinctiveness of ASD and ADHD. However, in our opinion, the results and lessons learned are limited in scope. The overall quality of the manuscript should be improved, for example by adding some information on the need and implications for both clinicians and researchers to disentangle the spectrum of
NDD, and by reviewing the state of research on the overlap/distinctiveness of ASD and ADHD.  References should also be enriched by more recent papers.

Methodological issues above-mentioned should be clearly discussed in the limitation part. Finally, it represents a preparatory work that leaves us expecting further research

Author Response

Dear Reviewer 2,

We are most grateful to you for the helpful comments on the original version of our manuscript entitled “The Relationship between Motor Coordination Ability, Cognitive Ability, and Academic Achievement in Japanese Children with Autism Spectrum Disorder and Attention Deficit/Hyperactivity Disorder”. We have taken all these comments into account and herewith submit revised version of our paper.

We hope that the revised version of our paper is now suitable for publication in Brain Sciences and we look forward to hearing from you at your earliest convenience.

Round 2

Reviewer 1 Report

The revised version of the manuscript considers and discusses all comments.  Clarifications are provided when requested